# Urban density and spatial planning: The unforeseen impacts of Dutch devolution

Jip Claassens[1☯]*, Eric Koomen[1☯], Jan Rouwendal[2☯]

**1** Department of Spatial Economics, Spatial Information Laboratory (SPINlab), Vrije Universiteit Amsterdam, Amsterdam, The Netherlands, **2** Department of Spatial Economics, Vrije Universiteit Amsterdam, Amsterdam, The Netherlands

☯ These authors contributed equally to this work.
* j.claassens@vu.nl

**Data Availability Statement:** All used source files are available from the PDOK.nl database: Building database from 2012 to 2018 (Basisregistratie Adressen en Gebouwen) http://geodata.nationaalgeoregister.nl/inspireadressen/extract/

## Abstract

National spatial planning has strongly influenced urban development and open space preservation in the Netherlands since the 1950s and established the country's reputation as a planner's paradise. The gradual withdrawal from this active and stringent type of planning in favour of decentralisation and deregulation has received less attention and its impacts on urban development patterns remain poorly studied. This study investigates residential development since 2000 in relation to the changing planning context. We focused on residential densification and the redevelopment of greyfields and brownfields as desired outcomes of policies that aim to limit urban development in open landscapes. Using detailed spatial housing and land-use data we quantified the relative importance of different urban development processes over three subsequent six-year periods characterised by slowly decreasing national policy attention to steering residential development. Our results indicate that, while the national policy instruments got weaker, the share of residential development within existing urban areas increased. Our results lend further credence to the suggestion that the abandoned national spatial planning policy targeted at housing construction within urban development zones that were predominantly defined on greenfield near existing cities, limited urban redevelopment. Despite reduced government spending, densities increased within existing urban areas as general, local-level restrictive policies with respect to greenfield development remained in place and demand for urban housing remained unabated.

## 1. Introduction

Cities are expanding worldwide. Regardless of whether these expansions occur via zoning plans or unbridled urban sprawl, cities have increased in surface area and open space is sacrificed for the ever-growing demand for urban space [1–3]. However, cities also grow by adding new houses to the existing urban fabric and thus increasing the initial density, a term we refer to as "densification".

Planners often express a preference for densification over spatial expansion. The promotion of building within existing urban contours dates back at least to the 'compact city' concept first

inspireadressen.zip Land-use maps 1996 to 2015 (Bestand Bodemgebruik) http://geodata. nationaalgeoregister.nl/elu/extract/elu.zip Urban contours in 2000 (Begrenzing Bebouwd Gebied) https://doi.org/10.34894/PFK0OB Building counts per hectare from 2000 to 2014 (CBS vierkanten 2014) http://geodata.nationaalgeoregister.nl/ cbsvierkanten100mv2/extract/ cbsvierkanten100m20002014v1.zip Residential development zones (Vinex locaties) https://doi.org/ 10.34894/DOJUJN Bundling zones (Bundelings gebieden) https://doi.org/10.34894/FYBGI0 Protected nature areas (Natura2000) https://www. pdok.nl/geo-services/-/article/beschermde-gebieden-natura2000-inspire-geharmoniseerd-#9cb3abe01423cd065a2411e5ef916ca7 Deprived neighbourhoods in 2007 (Locaties krachtwijken) https://doi.org/10.34894/SOITGE Travel time to railway stations 2006 https://doi.org/10.34894/ JG605M Travel time to 100,000 inhabitants in 2004 https://doi.org/10.34894/PXKGI4 Urban attractivity index in 2002 https://doi.org/10.34894/ AXCRCU.

**Funding:** For this study we have received financial funding from the Dutch Research Council (NWO) with grant number: 438.16.152. The funders had no role in study design, data collection and analysis, decision to publish, or preparation of the manuscript. Moreover, the authors did not receive a salary from our funders.

**Competing interests:** The authors have declared that no competing interests exist.

proposed by Dantzig and Saaty [4], partly as a reaction to the urban sprawl trends in the decades before. A compact city has been described as "*a relatively high-density, mixed-use city, based on an efficient public transport system and dimensions that encourage walking and cycling. It contrasts with the car-oriented 'urban sprawl' of many modern towns and cities.*" [5]. In North America, this *compact city* idea is closely related to *new urbanism* or s*mart growth* [6].

There are multiple advantages associated with densification as an urban development option. First, construction within the existing built-up area reduces pressure on the surrounding open space which is generally regarded as an important amenity [7]. Second, a higher density can help to limit automobile travel, as it may shorten travel distances to destinations and make more sustainable transport modes more viable [8, 9]. Third, it can act as an urban renewal instrument when it replaces outdated older structures or un(der)used sites by more attractive buildings, thereby increasing the overall quality of the urban area for society [5, 10]. Fourth, it may decrease household energy consumption, not only through less car traffic *per capita*, but also by diminishing heating energy demand, and improving opportunities for shared energy systems [5, 11, 12]. Fifth, housing development costs could be lower by utilising (and if needed upgrading) existing infrastructure more intensively. Lastly, higher density and compactness are associated with mixed land use, diversity, social cohesion, and cultural development [13].

It is not completely clear if, under which conditions and to what extent, all these perceived advantages of densification materialise in specific cases. What is clear is that incumbent residents often oppose densification of their neighbourhood, which strongly suggests that they anticipate negative effects. Densification may diminish the number or size of parks and public spaces, cause property depreciation, or result in a change in social composition, and hence threatens to decrease their quality of life [14]. Planning procedures often take opinions of incumbent inhabitants into account for any proposed change in land use and weigh them heavily. The degree to which spatial planning policies are able to steer residential development towards the desired degree of densification is therefore in general not easy to assess. The effect of opposition against it may easily be that planning procedures tend to 'freeze' land use in existing built-up areas where changes are difficult to implement. Prospects for dense development are thus better in greenfield areas that will be newly developed, but this implies an expansion of the urban area.

While densification is often proposed in spatial planning, it is unclear what the actual impact of planning policies on densification is. Usually, planning policies have a number of ambitions that may not be easy to realise simultaneously. For instance, the previous paragraph suggests that promoting the quality of life in urban neighbourhoods may not fit easily with densification. Planning goals interact with societal preferences and market forces and it is often not easy to isolate the impact of spatial planning policies on densification from the many other forces that are simultaneously relevant. The problem is that an undisputed, counterfactual reference point is lacking: in retrospect, it is not easy to determine what would have happened in the absence of a policy. Changes in policies, however, are potentially informative in this respect. For instance, if an existing planning measure is discontinued or a new one introduced, observing what happens before and after such an event may provide valuable insights into its effects and thus inform planners of possibly effective policies.

This paper analyses such a change in the Netherlands, where active national spatial planning with respect to housing construction has been fading since the turn of the century in favour of decentralised planning alongside a continuation of increasing possibilities for market-driven initiatives [9, 15–17]. This so-called devolution process is often related to the neoliberal emphasis on limiting government intervention and can be observed in several European countries, most notably the United Kingdom [17–19].

This shift in spatially-explicit national planning policy is a significant one, but its impact appears to have hardly been studied. In this paper, we consider in detail the urban densification of housing in the Netherlands during the period in which this gradual policy shift took place. Our research objective is to assess the impact of changes in planning policies on the amount and location of densification in the Netherlands between 2000 and 2018. To our knowledge, this has not been done before, at this level of detail and scope. Our approach builds on a rich set of spatial data sources to identify the evolution of the number of houses per hectare in the period 2000 up to and including 2017. Using detailed spatial housing and land-use data we quantified the relative importance of different urban development processes over three subsequent six-year periods characterised by slowly decreasing national policy attention to steering residential development.

This paper is structured as follows: first, we provide more discussion of the Dutch context, then in section 3 we introduce the data and methods, which are divided into two parts. After looking at the locations and quantity of development, we try to statistically explain why development happens at those locations and in those quantities. In section 4 we present the findings of the research. The final section summarises, draws some conclusions and suggests issues for future research.

## 2. Dutch context

### 2.1. Study area

The Netherlands is among the most densely populated countries in the world [20]. Its 17 million inhabitants are spread over a relatively small area, but the largest city, Amsterdam, has less than one million inhabitants. Over 80% of the country's population lives in (sub)urban areas that are relatively well connected [21]. The country has experienced rapid population growth until the 1960s and unlike some other European countries, its population is still growing. For the coming decades, this trend is expected to continue. Recent projections in the Netherlands indicate the need for 300,000 to 1,600,000 new housing units before 2050 [22–24] to accommodate future growth of the population–mainly through migration–as well as diminishing household sizes [25]. This expected demand has initiated a lively debate on where new houses should be built and to what extent the required growth of the housing stock can be realised within existing urban areas. Existing projections range widely between 9%-80% [24, 26]. Understanding current residential development dynamics and their relation to spatial planning initiatives can help to draft realistic and effective policy alternatives.

### 2.2. Planning policies in the Netherlands

During the second half of the 20[th] century, the Netherlands developed a strong spatial planning tradition that was characterised by relatively strict spatial policies defined at the national level that limited urban development in some areas (e.g. buffer zones) and stimulated it in others (e.g. new towns and spatial development zones). An extensive account of these policies and their effectiveness is provided elsewhere (e.g. [27, 28]). From the 1980s spatial planning focussed on establishing compact cities, stimulating densification of existing urban areas and designating development zones within and nearby cities [15, 29, 30]. These zones were not exclusively top-down defined, but rather developed in a "consensus-seeking process which rested on the assumption that local and provincial authorities were willing to cooperate" [15], and then allocated national funds were part of this negotiated deal [15]. Since 2004, national government maintained its overall objective of compact urban development and thus densification, but gradually changed its approach to spatial planning by abolishing nationally designated spatially-explicit planning concepts (e.g. buffer zones, spatial development zones) and

handing over more responsibilities to lower levels of government. This devolution process provided more possibilities for municipalities to take the initiative for developing spatial visions and development plans [31–34]. Following this gradual government retreat from spatially-explicit zoning plans (which started in 2001), a more generic development guideline was implemented in 2012. This stepwise approach prescribes development within existing urban areas, except when this is not realistically possible. In this case, limited development is permitted outside urban areas [35].

Land-use planning can be passive, by prohibiting some developments and permitting others, or active by taking actions that lead to the realisation of the desired development. Active land-use planning requires that the government takes (financial and other) responsibility, and the national government did so at least until the 1990s and early 2000s. Formally, land-use planning in the Netherlands has a hierarchical structure, but in practice, negotiations were the main instrument used by the national government to influence the plans of the lower levels, and making money available for the desired developments. In practice, provinces always had a less influential position, mainly transferring national initiatives, regulations and funds to the local level. According to Faludi and van der Valk [36] provinces are the 'lynchpin' of the Dutch planning system. One argument for this is that they have regulations that prohibit building outside the built-up areas and do not lift them easily. This is a passive planning instrument. To realize new residential development active planning is needed and here the provinces are in a much weaker position. Municipalities are a key player in Dutch land-use planning because the local land-use plans, which determine which developments are permitted at a particular site, are usually determined at this level. However, since 2008 the national and provincial governments can impose local land-use plans and overrule municipalities, but this rarely happens (see Needham, 37). Active land-use planning at the local level means, first of all, that the local land-use plan should facilitate the desired developments and second that the municipality takes the required actions. Until 1994 municipalities usually took all responsibility for residential development enabled by the financial means provided by the national government. With these funds getting scarcer, partnerships with commercial developers became more common and their role has grown over time (see Needham, [37]).

The 'Nota Volkshuisvesting in de Jaren Negentig' from 1989, was an important tipping point from where there was to become less government and more market [3]. Until 1995, the national government bore all the financial risks associated with large scale residential development [3]. This changed in 1995 when the national government signed contracts with other governmental bodies about their spatial planning ambitions in which they agreed to target urbanisation in 26 so-called *VINEX*-locations—designated expansion locations or urban regions—and determined the number of houses for each of them [38]. Although the national government still provided location-specific subsidies for the realisation of these plans, in contrast to earlier practice it did so now as a lump sum for residential construction, leaving all the financial risks for the local governments.

After 2004, the government subsidies for residential construction gradually diminished to zero. Among those was the investment budget urban renewal (in Dutch referred to as ISV) that amounted 1.8 billion euro in 2000–2004 [39], 1.05 billion in 2005–2009 [40], 0.85 billion in 2010–2014 [41] and that was absent afterwards. Two other subsidy sources focussed on the development of residences and locations (BWS and BLS, respectively) and were discontinued in 2005 [42].

Moreover, from 2004 onwards the government left the initiative for determining the location and size of the residential construction plans to municipalities, which often set up partnerships with private parties, usually large developers. The change in the policy thus appears to have provided more room for market-driven development initiatives [33].

It is important to note that the growth of the housing stock has not been constant over time. Whereas the net increase was around the 50,000 per year just after the war, it increased to 140,000 per year in the seventies and gradually decreased later on. Since 2000 annual housing construction has been relatively stable at around 60,000 housing units per year despite the withdrawal of subsidies from the national government that were perceived as a necessary condition for housing construction in earlier decades [43].

## 3. Data and methods

We discuss the various types of residential development processes in Section 3.1. The subsequent section introduces the spatial analysis that applies highly detailed data sources to characterise the local amount of residential development per process for three consecutive six-year periods of declining importance of national spatial planning. The statistical methods used to explain the location of development and their links with the gradual shift in spatial policy are described in Section 3.3.

### 3.1. Classifying residential development processes

In this paper, we focused on the number of housing units per hectare and described residential development accordingly as an increase in the number of housing units per hectare. Within existing urban areas, we considered all (re)development of the built environment that results in higher densities per hectare as densification [10, 44, 45]. More specifically, we considered four different types of densification within urban areas: *brownfield redevelopment*, *greyfield redevelopment*, *residential densification*, and *urban greenfield development*.

Brownfields are defined by the US Environmental Protection Agency [46] as "*abandoned, idle, or under-used industrial and commercial properties where expansion or redevelopment is complicated by real or perceived environmental contamination*". Brownfields represent a potentially lucrative, but largely untapped, land resource because of their size and location [47, 48]. Closely related is greyfield redevelopment. These areas are similar to brownfields but lack their (perceived) environmental contamination [49], and typically refer to offices, shops and paved-over areas such as parking lots. Brownfield and greyfield development usually have a large impact on the physical appearance of the concerned neighbourhoods. The third type of development is densification through the addition of extra housing units to areas that are already predominantly residential. This type of densification can be subdivided into two categories: *hard* and *soft densification* [14], where *hard densification* refers mainly to the demolition of existing buildings and their replacement by new housing units [14, 50]. Like brownfield and greyfield development it may completely change the appearance of neighbourhoods. *Soft densification* is accomplished by, for example, adding extra floors to existing buildings [50], or by building extra housing units on existing larger residential parcels that were still vacant [14, 51], or by splitting existing larger units into smaller units [50]. This type of densification often has a much smaller impact on urban structure and sometimes is hardly visible to a visitor of the area. The last type is urban greenfield development and it relates to the construction of new housing units on urban green areas such as sports facilities, allotment gardens, and parks [52].

Residential development outside larger contiguous urban areas is usually greenfield development of "*clean agricultural or open land sites*" [53], and thus refers to agricultural lands, natural areas, and open water. Although this obviously comprises the largest share of development in these predominantly non-urban areas, occasionally small patches of residential or industrial land use that are too small to be considered contiguous urban areas can also be found here. For simplicity, we used the same classification of development processes as for

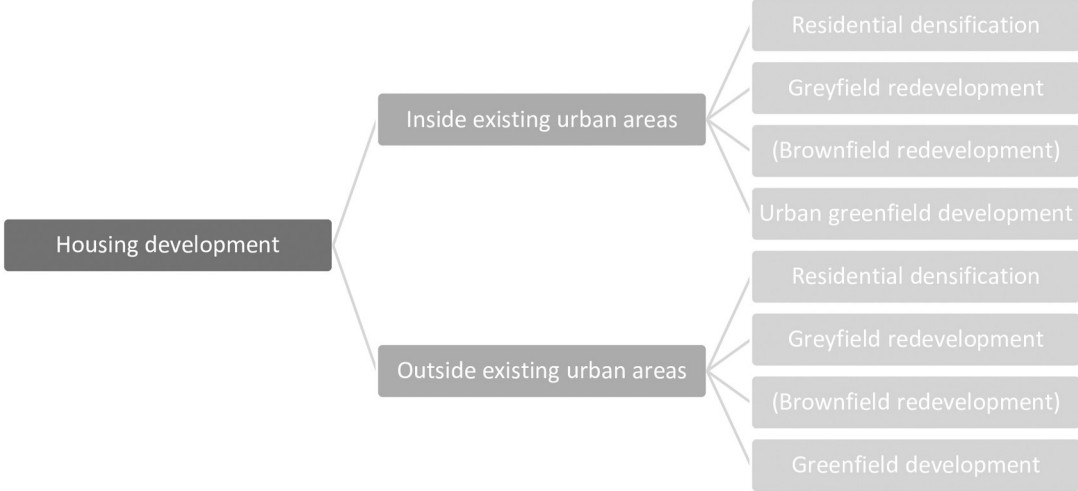

**Fig 1. Classification of residential development processes.** Brownfield redevelopment is combined with greyfield redevelopment in this particular study due to data limitations.

inside urban areas. A graphical overview of the various processes that contribute to residential development is included in Fig 1.

## 3.2. Identifying residential development processes since 2000

To investigate residential development as defined and classified above in the Netherlands, we used highly-detailed cadastral information that describes the location, age and function of all buildings in the country. Based on this large dataset we counted the number of residences in all 100x100 metre (hectare) cells at the start of 2012 and 2018. These two cross-sections are complemented with a data set from Statistics Netherlands (CBS) that contains the number of residences for the same 100x100 metre cells in grids dating back to 2000. The total number of housing units in both data sets differ 5% for the overlapping year due to differences in definitions in the underlying housing registration databases. These differences do not influence our analysis as they are homogenously distributed over the country. We furthermore defined our analysis periods such that they are based on a single data source. This information allowed us to compute the change in the number of housing units per hectare for three consecutive six-year periods of the declining importance of national spatial planning: 2000–2005; 2006–2011; and 2012–2017. The first period coincides with the last years in which the policy measures proposed in the Fourth Memorandum on Spatial Planning were carried out. That Memorandum defined spatially-explicit residential development zones as large-scale building sites close to the central cities [54, 55]. The second period roughly matches the years of the succeeding national spatial planning document [56] that emphasised a new steering philosophy—development planning in which national government supports other tiers of government and private stakeholders–to replace former restrictive planning [57]. The final period overlaps with the current prevailing national spatial planning document [58] that leaves the initiative for spatial development to local governments and private parties, abolishing, for example, the national buffer zones in which urban development was discouraged and the bundling zones in which residential development was hitherto preferably concentrated [28, 35]. Obviously, planning initiatives may have had impact beyond the exact period in which they were defined. So, we viewed the analysis periods as being indicative of the general transition from fairly directive

national planning guidelines with spatially-explicit ambitions, to more decentralised conditions in which market forces were expected to take over.

The observed local changes in housing units were classified into one of the main development processes discussed in Subsection 3.1 and depicted in Fig 1, based on their initial land use, in 2000. This was inferred from a spatial overlay with a dataset describing land use in 2000 produced by Statistics Netherlands [59]. This data set contains 38 types of use that are aggregated into main land-use classes relevant to the distinguished processes: residential areas, greyfields (due to data limitations, we combine brownfields with greyfields), and green areas (S1 Table). We thus classified development relative to land use in 2000. For areas classified as building lots in 2000, we referred back to their previous use in an earlier version of the land-use dataset. In addition, we applied the contours of contiguous urban areas for the year 2000 developed by the Dutch government which are based on a spatial analysis of the same land-use data [60]. Hence, our distinction between developments within and outside existing urban areas always refers to the contours of 2000. The former developments contribute to the further intensification of initial urban areas, whereas the latter are considered as urban extensions in formerly non-urban areas. The information about the changes in residential units was then combined with the 6-class land-use map of 2000, to find changes per development class. Note that this change in the number of housing units is a net result, based on the number of newly-added minus withdrawn housing units per grid cell. It sometimes also happens that the number of housing units decreases locally, however, this occurs relatively infrequently and is usually a precursor of redevelopment and a consequence of our selection of periods. In the explanatory analysis, we, therefore, omitted net negative developments.

## 3.3. Explaining residential development processes

To delve further into the implications of the policy shift, we set up a binomial logistic regression and a Poisson regression in a panel set-up to investigate which spatial factors are associated with the probability of a cell being developed for housing or the amount of development in that cell. The main purpose of this analysis is to investigate the role of spatial policy. We do so by introducing a number of variables indicating that a grid cell is located in an area designated for specific land-use policy targets. A second important purpose is to investigate if the shift to decentralised and market-driven developments is reflected in a more prominent role of the variables commonly associated with urban development in the course of the whole period considered here.

**3.3.1. Explanatory variables.** To realise the first target, we collected several data sets reflecting spatially-explicit policies that were in place during the study period. Like the other data sets included in this study, they are processed to 100x100m sized raster cells. The first and most important is an indicator whether a cell is located in one of the previously mentioned *residential development zones* or *VINEX* which are locations that were designated in 1991 to accommodate residential growth under the then prevailing national land-use policy. They were largely–but not exclusively—located outside the existing urban fabric, and therefore mainly linked to expansion [61]. This is the variable of main interest in this study and we expect it to be positively related to the number of new houses realised, especially in the first period.

Several other land-use measures were in force during the periods considered in this research. These policies may also have affected residential development and it is, therefore, important to control for them to get a proper view on the impact of the fading VINEX policy. A second important concept from the prevailing national planning memoranda in the studied period are *bundling zones*. These are existing clusters of urban areas wherein development

should be concentrated in order to fully utilise existing investments in infrastructure and services [56]. That policy was implemented in 2004 and discontinued in 2012. We, therefore, expect its impact to be especially significant in the second period, 2006–2012.

Thirdly, densification may be related to a place-based policy targeted at *deprived neighbourhoods*. 40 such neighbourhoods were designated by the government in 2007 and redevelopment subsidies were provided [62].

Land-use policy has a close connection to environmental policy and it is important to control for the possible impact of it on densification. *Designated natural areas* are Europe's protected *Natura2000* areas where no urban development is allowed and are therefore expected to have a low probability that new houses are added throughout the period 2000–2017. Since Natura2000 areas are not located in areas that were urban in the year 2000, we will only include it for those processes outside the urban contour.

Accessibility is perhaps the most important variable that is commonly related to urban development [63–65]. In this study, we used two distance variables as proxies for centrality, distinguishing between local and regional centrality. For local accessibility, we used a network-based travel time to 2001-railway stations. The choice for railway stations is twofold; railway stations are often centrally located in cities or villages and act therefore as a proxy for the city centre, and secondly, many workers use the train to get to work, and therefore, living close to railway stations may be preferable. In fact, proximity to railway stations appears to be more important for residential location choice than highway exits in the Netherlands [63]. So we did not include the commonly-used proximity to highway exits. To emphasise the local character of this measure, we used a cut-off value for travel time at 15 minutes. This value was chosen because 66% of train travellers in the Netherlands in 2014 cycled or walked from home to the train station [66]. For regional accessibility, we used a network-based travel time to reach 100,000 inhabitants. So for each location in the country, we calculated the shortest possible time needed to travel over the road network to reach 100,000 inhabitants [64, 67]. This variable covers the regional scale and identifies more populated clusters of towns and cities. We limited the maximum travel time to 120 minutes to omit the outliers (e.g. the northern Dutch islands only accessible by boat). While accessibility is traditionally important for new urban development, it would be interesting to see whether this also explains densification and to what degree.

In addition to accessibility, location characteristics are often regarded as a driver of urban development [64, 65, 68, 69]. Amenities are the most prominent example [70–73] Therefore, we used an *Urban attractivity index*, which is an index we developed to represent the weighted and standardised number of amenities in 2002 in a 500 meter grid cell and averaged out over a 2.5 km radius. It is comprised of historic buildings and monuments, cultural facilities, shops, hotels, restaurants, and other catering establishments.

We were also interested in the effect of the initial housing density at a location on redevelopment and included therefore a variable with the housing density in 2000. Descriptive statistics for these variables can be found in Table 1.

**3.3.2. Regression framework.** We then used these variables in a panel regression framework comprising of two parts. First, we explain the occurrence of development, which is defined as 1 if development is larger than 2 (due to rounding in the datasets of the first two periods), using binomial logistic regression. Whereas the second part explains the amount of development using Poisson regression since this data can be regarded as count data. The binomial logistic regression analysis takes the following simplified form:

$$D_{it} = I(\alpha + \beta Z_i + \delta D_i + \lambda L_i + u + \varepsilon > 0) \qquad \text{Eq1}$$

**Table 1. Descriptive statistics for cells with urban residential area.**

| Variable | Obs. | Mean | Std. Dev. | Min | Max |
|---|---|---|---|---|---|
| *Planning policy factors* | | | | | |
| Residential development zone present | 3,520,251 | 0.005 | 0.068 | 0 | 1 |
| Bundling zone present | 3,520,251 | 0.136 | 0.342 | 0 | 1 |
| Designated natural area present | 3,520,251 | 0.103 | 0.304 | 0 | 1 |
| Deprived neighbourhood present | 3,520,251 | 0.003 | 0.058 | 0 | 1 |
| *Accessibility factors* | | | | | |
| Network distance to nearest railway station (in min) | 3,520,251 | 9.384 | 4.099 | 1 | 15 |
| Network distance to nearest 100,000th inhabitant (in min) | 3,520,251 | 46.611 | 21.556 | 4 | 120 |
| *Location factors* | | | | | |
| Urban Attractivity Index | 3,520,251 | 0.529 | 3.273 | 0 | 87 |
| Housing density in 2000 (per 10 housing units) | 3,520,251 | 0.180 | 0.864 | 0 | 48.5 |

where $D_{it}$ is a Boolean (zero-one) variable indicating that the number of housing units increased in cell i in period t. $I(.)$ Is the indicator function that equals 1 if the inequality within the brackets is satisfied and 0 otherwise. The left-hand side of the inequality can be regarded as a latent variable reflecting the propensity to add houses. $Z_i$ is a vector of planning policy factors, $D_i$ is a vector of accessibility factors, and $L_i$ a vector of location factors and u is the random effects estimator.

In this research, we focused on densification and therefore we ignored the local decline in the number of housing units per hectare because this pertains to a relatively small share of the changed cells (16% for the period 2000–2006), and this typically signals the demolition of housing preceding new construction in the subsequent period. This approach may cause an overestimation of overall densification, when ignoring the local decline in densities, but is expected to be limited.

The Poisson regression uses the number of houses added per cell ($A_{it}$) as the dependent variable and applies the same explanatory variables as described above:

$$A_{it} = \alpha + \beta Z_i + \delta D_i + \lambda L_i + u + \varepsilon \qquad \text{Eq2}$$

## 4. Results

### 4.1. Identifying residential development processes

The results of this analysis are summarised in Table 2. The number of housing units added indicates the net changed, computed as the number of newly-added minus withdrawn housing units per grid cell, and only includes cells where this difference is positive. Recall that the number of cells with a net negative change is small, and that such a change usually predicts redevelopment at a larger density in the following years. The overestimation of the number of houses added implied by ignoring the cell with a negative net change is negligible. In the studied periods around one million housing units were added. Interestingly, the total amount of new housing per period did not change substantially over time when the national construction programs on designated sites gradually faded out. This is our first important conclusion: the disappearance of national coordination and the abolition of development subsidies did not result in a drop in the number of housing units added to the stock.

Table 2 also shows a large reduction in the share of new housing constructed outside existing urban areas: from 58% in the first period to 31% in the third. To interpret this change properly it should be born in mind that many of the greenfield areas outside cities designated for housing construction under the initial national planning regime were still under

**Table 2. Residential development patterns between 2000–2005, 2006–2011, and 2012–2017.**

| 2000–2005 | Development process | Housing units added | Affected cells | Share per process | Share urban /non-urban | Density* |
|---|---|---|---|---|---|---|
| Urban Area | Residential densification | 79,900 | 14,888 | 25% | 42% | 32.8 |
| | Greyfield development | 44,195 | 4,326 | 14% | | 33.1 |
| | Urban greenfield dev. | 12,595 | 1,435 | 4% | | 21.7 |
| Outside Urban Area | Residential densification | 16,810 | 1,146 | 5% | 58% | 17.7 |
| | Greyfield development | 29,370 | 1,981 | 9% | | 17.9 |
| | Greenfield development | 141,235 | 10,397 | 44% | | 15.6 |
| | Total | 324,105 | 34,173 | 100% | 100% | |
| **2006–2011** | | | | | | |
| Urban Area | Residential densification | 74,110 | 16,711 | 21% | 47% | 38.2 |
| | Greyfield development | 68,205 | 5,778 | 19% | | 36.8 |
| | Urban greenfield dev. | 21,660 | 1,785 | 6% | | 26.9 |
| Outside Urban Area | Residential densification | 3,275 | 511 | 1% | 53% | 14.5 |
| | Greyfield development | 22,615 | 2,050 | 6% | | 16.7 |
| | Greenfield development | 160,085 | 13,440 | 46% | | 15.7 |
| | Total | 349,950 | 40,275 | 100% | 100% | |
| **2012–2017** | | | | | | |
| Urban Area | Residential densification | 88,361 | 13,620 | 27% | 69% | 36.9 |
| | Greyfield development | 112,000 | 6,538 | 34% | | 38.4 |
| | Urban greenfield dev. | 22,674 | 1,610 | 7% | | 24.9 |
| Outside Urban Area | Residential densification | 771 | 184 | 0% | 31% | 12 |
| | Greyfield development | 2,999 | 2,948 | 1% | | 13 |
| | Greenfield development | 98,031 | 8,953 | 30% | | 12.5 |
| | Total | 324,836 | 33,853 | 100% | 100% | |

*Average density in housing units per hectare of those cells where houses were added.

construction during the second and third period. That is, a substantial share of the greenfield development outside urban areas in the second and third sub-periods is still related to plans prepared under the earlier planning regime with national coordination, which implies that we cannot yet see the full impact of the policy change. However, our second main conclusion is that the substantial drop in residential development outside existing urban areas, which was realised especially in the third period, was not associated with a drop in the total addition of new housing units. Clearly, more housing units were added inside the existing urban area and especially in the years 2012–2017.

The table indeed highlights that approximately 42% of all the housing units added between 2000 and 2005 were located within existing urban areas and that this figure increases to approximately 69% between 2012 and 2017. What makes this is even more surprising is that urban land use in 2000 has been taken as the reference for all three periods and one could expect that it became more difficult to add housing units when time progressed and the most obvious locations for (re)development were filled. When we change the urban contours to that of the beginning of each period, as a sensitivity analysis, that figure goes from 42% to 82% in the last period. Our third important conclusion is that most of the increase in intra-urban densification is due to urban greyfield (including brownfield) development and–at a much smaller scale–urban greenfield development, while the share of residential densification changed only modestly. The share of greyfield development more than doubled from 14% in 2000–2005, to 19% in 2006–2011 and 34% in 2011–2017. The share of urban greenfield development is small

throughout the period, but it also increased considerably, from 4% to 7%, that is by more than 50%. In contrast, residential densification changed only moderately, by less than 20%.

Table 2 also shows the average density in housing units per hectare of those cells where houses were added, and it shows that the density within urban areas is much higher than outside them. This is also displayed in the average size of the housing units within urban areas, on average 100 m$^2$, and 130 m$^2$ outside urban areas (calculated for 2017 in the areas that changed). Moreover, the density outside urban areas is decreasing, while the density in urban greyfields is increasing.

The findings, listed in the previous subsection, are surprising. How can it be that the fading of a national policy and associated subsidies aiming at densification did not result in less but more actual densification? An obvious possibility is that the abolition of large-scale plans for new residential quarters constructed mainly on greenfields contributed to pressure on the intra-urban areas through planning or market forces. This would be natural to expect unless local governments relaxed restrictions on greenfield development, for instance by developing plans themselves. There are indeed no clear signs that local or provincial authorities did this. It may be the case that provinces are more reluctant to facilitate greenfield development proposed by municipalities in more recent years, than the greenfield developments proposed in the national plans in earlier years. However, we are not aware of increased tension between municipal and provincial planning policies. By fading out its centralised planning system for housing construction, Dutch land-use policy moves into the direction of the U.K. system where proposals for development need to pass local public agencies, a process that according to many observers has contributed significantly to housing shortages and high house prices (see for instance Hilber and Vermeulen [74]. House prices increased by more than 50% in the Netherlands over the period considered [75, 76], although there was a temporary drop associated with the euro crisis. These higher house prices have probably contributed to the attractiveness of developing urban greyfields–often areas abandoned by manufacturing industries or public utilities–into residential areas. The same incentive works for open space inside urban areas. For residential areas, the incentive for redevelopment is at least partly muted by the fact that higher house prices also increase the value of existing buildings, not only that of the new ones.

The result just presented suggests that despite its impeccable intentions, the active national land-use and housing-construction policy prevented the realisation of densification inside existing urban areas. The reason is that it realised large-scale greenfield development close to existing urban areas, and thus functioned as a valve to relieve demand pressure there. The fading out of this policy implied the closing of the valve and provided a powerful incentive for more densification inside existing areas. To be sure, this interpretation needs further elaboration. It is consistent with the four conclusions formulated above, but the mechanism it supposes cannot be studied with the data used in this study.

## 4.2. Explaining residential development processes

In what follows we report estimation results for the three dominant development processes, which together comprise more than 80% of the newly-added units: *urban residential densification*, *urban transformation*, and *greenfield development outside urban areas*.

In Table 3 regression results for the three main development processes are presented. The first three columns show the coefficients for the binomial logistic regression explaining whether or not development occurred. While the latter three columns show the coefficients for the Poisson regression explaining the amount of change. Specification 1 shows the results for a logit specification where the chance of developing a hectare within the first process (urban

**Table 3. Regression results for the three main development processes, for the periods only.**

|  | (1) | (2) | (3) | (4) | (5) | (6) |
|---|---|---|---|---|---|---|
|  | Urban densification | Urban greyfield | Non-urban greenfield | Urban densification | Urban greyfield | Non-urban greenfield |
| VARIABLES | Logit | Logit | Logit | Poisson | Poisson | Poisson |
| 2000–2006 | -3.369*** | -4.803*** | -8.887*** | -0.474*** | -0.791*** | -3.024*** |
|  | (0.015) | (0.035) | (0.025) | (0.009) | (0.017) | (0.009) |
| 2006–2012 | -3.294*** | -4.438*** | -8.608*** | -0.429*** | -0.378*** | -2.888*** |
|  | (0.015) | (0.032) | (0.024) | (0.009) | (0.017) | (0.009) |
| 2012–2018 | -3.452*** | -4.196*** | -9.097*** | -0.292*** | 0.160*** | -3.163*** |
|  | (0.016) | (0.031) | (0.026) | (0.009) | (0.016) | (0.010) |
| Observations | 625,275 | 342,123 | 8,943,552 | 625,275 | 342,123 | 8,943,552 |
| Number of groups | 208,425 | 114,041 | 2,981,184 | 208,425 | 114,041 | 2,981,184 |

Standard errors in parentheses

*** p<0.01

** p<0.05

* p<0.1

densification) is estimated. Converting the logit coefficients to probabilities gives a probability of 0.033, 0.036, and 0.031 respectively for urban densification in the three periods. This indicates that the probability that development occurs in a residential area within the urban contour of 2000 remained fairly constant over the years. However, for urban greyfield development, this probability increased from 0.008 to 0.015. The probabilities are very low due to the many cells where nothing happened. In specification 4–6 the Poisson regression results are showed. When we exponentiate the Poisson coefficients we get the average amount of development per hectare, if we then multiply this by the number of cells considered (208,429 for urban densification), we get the total amount of development in that period for that process. For densification in the first period, this is 129,749 which is very close to the actual value of 129,740 (see S2 Table). Note that this incorporates only positive development, hence this value is a bit different than the number of housing units added in Table 2.

With respect to urban densification, column (1) in Table 3 shows that the probability that at least one unit was added to a cell increased slightly in the second period, but decreased in the third one. Column (4) implies that the expected number of units added per cell increased slightly in the second period, but substantially in the third one. Indeed the increase in the expected number of units compensated completely for the smaller number of cells with a net addition to the housing stock. A fifth important conclusion is that It appears that the urban densification in more recent years was realised not by adding one or a few units more or less randomly over space, but through the realisation of larger numbers of dwellings, for instance by replacing single-family housing by apartment buildings. Column (5) shows that urban greyfield development has shown an even larger increase in the number of dwellings added per cell over time.

Next, we want to explore the effects of policy measures over time. We, therefore, interact these policy measures with the three periods (Table 4). From these results, we see a decreasing effect of VINEX locations on the probability of development in urban residential areas 0.90 to 0.28 (transformed logit coefficients). Furthermore, we also see that the amount of development per hectare in VINEX areas decreases over time. A similar trend, albeit with a smaller decrease, is visible for development in urban greyfield that are VINEX locations. For VINEX locations on non-urban greenfields, the probability of development decreased but remained very high.

**Table 4. Regression results for the three main development processes, with policy measures and control variables.**

| VARIABLES | (1) Urban densification Logit | (2) Urban greyfield Logit | (3) Non-urban greenfield Logit | (4) Urban densification Poisson | (5) Urban greyfield Poisson | (6) Non-urban greenfield Poisson |
|---|---|---|---|---|---|---|
| 2000–2006 | -2.994*** | -4.887*** | -6.896*** | -0.309*** | -1.393*** | -2.350*** |
| | (0.044) | (0.080) | (0.055) | (0.034) | (0.059) | (0.027) |
| 2006–2012 | -3.206*** | -4.387*** | -6.376*** | -0.447*** | -0.747*** | -1.877*** |
| | (0.043) | (0.070) | (0.048) | (0.034) | (0.058) | (0.027) |
| 2012–2018 | -3.493*** | -4.257*** | -6.306*** | -0.369*** | -0.100* | -1.726*** |
| | (0.048) | (0.070) | (0.056) | (0.034) | (0.057) | (0.027) |
| Vinex # 2000–2006 | 2.235*** | 2.919*** | 5.602*** | 2.107*** | 2.812*** | 4.268*** |
| | (0.057) | (0.160) | (0.046) | (0.077) | (0.247) | (0.121) |
| Vinex # 2006–2012 | -0.218* | 1.085*** | 4.927*** | -0.456*** | 1.055*** | 3.654*** |
| | (0.113) | (0.229) | (0.044) | (0.084) | (0.251) | (0.121) |
| Vinex # 2012–2018 | -0.921*** | 0.943*** | 3.963*** | -1.025*** | 0.780*** | 2.901*** |
| | (0.169) | (0.215) | (0.050) | (0.089) | (0.249) | (0.121) |
| Bundling zone # 2000–2006 | -0.189*** | -0.401*** | 0.667*** | -0.009 | -0.126*** | 0.819*** |
| | (0.029) | (0.053) | (0.034) | (0.024) | (0.043) | (0.029) |
| Bundling zone # 2006–2012 | -0.128*** | -0.555*** | 0.680*** | 0.025 | -0.240*** | 0.850*** |
| | (0.028) | (0.047) | (0.030) | (0.024) | (0.042) | (0.029) |
| Bundling zone # 2012–2018 | -0.096*** | -0.125*** | 0.449*** | 0.002 | -0.064 | 0.572*** |
| | (0.031) | (0.045) | (0.035) | (0.024) | (0.042) | (0.029) |
| Depr. Neighbourhood # 2000–2006 | 0.741*** | | | 1.096*** | | |
| | (0.061) | | | (0.055) | | |
| Depr. Neighbourhood # 2006–2012 | 0.572*** | | | 0.995*** | | |
| | (0.053) | | | (0.055) | | |
| Depr. Neighbourhood # 2012–2018 | 0.571*** | | | 0.899*** | | |
| | (0.052) | | | (0.055) | | |
| Natura 2000 # 2000–2006 | | | -2.934*** | | | -3.275*** |
| | | | (0.166) | | | (0.070) |
| Natura 2000 # 2006–2012 | | | -3.157*** | | | -3.508*** |
| | | | (0.153) | | | (0.067) |
| Natura 2000 # 2012–2018 | | | -3.565*** | | | -2.976*** |
| | | | (0.227) | | | (0.056) |
| Initial density # 2000–2006 | -0.216*** | 0.189*** | | -0.184*** | 0.301*** | |
| | (0.006) | (0.009) | | (0.003) | (0.016) | |
| Initial density # 2006–2012 | -0.058*** | 0.232*** | | -0.022*** | 0.291*** | |
| | (0.005) | (0.008) | | (0.003) | (0.016) | |
| Initial density # 2012–2018 | -0.002 | 0.237*** | | -0.010*** | 0.293*** | |
| | (0.004) | (0.005) | | (0.003) | (0.016) | |
| Distance train station # 2000–2006 | 0.008*** | -0.002 | -0.034*** | 0.005* | -0.010** | -0.023*** |
| | (0.003) | (0.006) | (0.004) | (0.002) | (0.004) | (0.002) |
| Distance train station # 2006–2012 | -0.004 | -0.019*** | -0.043*** | -0.010*** | -0.027*** | -0.030*** |
| | (0.003) | (0.005) | (0.003) | (0.002) | (0.004) | (0.002) |
| Distance train station # 2012–2018 | -0.003 | -0.034*** | -0.027*** | -0.009*** | -0.027*** | -0.031*** |
| | (0.003) | (0.005) | (0.004) | (0.002) | (0.004) | (0.002) |
| Distance 100k inh. # 2000–2006 | -0.004*** | 0.003 | -0.032*** | -0.010*** | -0.011*** | -0.031*** |

*(Continued)*

**Table 4.** (Continued)

| VARIABLES | (1) Urban densification Logit | (2) Urban greyfield Logit | (3) Non-urban greenfield Logit | (4) Urban densification Poisson | (5) Urban greyfield Poisson | (6) Non-urban greenfield Poisson |
|---|---|---|---|---|---|---|
| | (0.001) | (0.002) | (0.001) | (0.001) | (0.001) | (0.001) |
| Distance 100k inh. # 2006–2012 | -0.002** | 0.004** | -0.033*** | -0.009*** | -0.010*** | -0.033*** |
| | (0.001) | (0.002) | (0.001) | (0.001) | (0.001) | (0.001) |
| Distance 100k inh. # 2012–2018 | -0.006*** | 0.001 | -0.047*** | -0.009*** | -0.018*** | -0.035*** |
| | (0.001) | (0.002) | (0.001) | (0.001) | (0.001) | (0.001) |
| Urban Attr Index # 2000–2006 | 0.056*** | 0.063*** | 0.125*** | 0.058*** | 0.085*** | 0.260*** |
| | (0.001) | (0.001) | (0.005) | (0.001) | (0.002) | (0.011) |
| Urban Attr. Index # 2006–2012 | 0.054*** | 0.061*** | 0.119*** | 0.048*** | 0.080*** | 0.262*** |
| | (0.001) | (0.001) | (0.005) | (0.001) | (0.002) | (0.011) |
| Urban Attr. Index # 2012–2018 | 0.060*** | 0.066*** | 0.117*** | 0.053*** | 0.077*** | 0.241*** |
| | (0.001) | (0.001) | (0.005) | (0.001) | (0.002) | (0.011) |
| Observations | 625,275 | 342,123 | 8,943,552 | 625,275 | 342,123 | 8,943,552 |
| Number of groups | 208,425 | 114,041 | 2,981,184 | 208,425 | 114,041 | 2,981,184 |

Standard errors in parentheses

*** p<0.01

** p<0.05

* p<0.1

This indicates that even in the last period those locations are not yet completely filled, but development per hectare was much lower.

The bundling zones were implemented in 2004 and discontinued in 2012. And interestingly, we see increasing significant results in the first two periods for urban residential and urban greyfield development, and insignificant coefficients for the last period. This matches well with the implementation and discontinuation of the policy.

The deprived neighbourhoods variable is only included for urban densification since this only applies to existing residential areas. The policy was implemented after 2007, and we would, therefore, expect a higher coefficient in the second and third period. However, the highest probability is in the first period indicating that these areas already showed, in fact, more development before this policy was enforced. But we will not pursue this intriguing observation here because it is outside the scope of the present paper.

The initial housing density in 2000 shows interesting results. For urban residential densification areas, if the housing density was higher in 2000 the odds ratio (exponentiated coefficient) of development is less than 1 (0.806) but it increases over time. Additional development in already dense areas was less probable all periods but the impact of existing housing decreased markedly over time. Moreover, the expected amount of development per hectare was also increased over time. This could mean that initially, densification happened predominantly in less dense areas, whereas later also areas that were already densely built-up were involved. For urban greyfields, development occurs more in initially dense areas and this increases over time, whereas the amount of development per hectare is constant over time.

The coefficients for designated natural areas for non-urban greenfields are large and negative, which means that the nature restriction was enforced. Whereas this variable is omitted for the urban residential densification and urban greyfield processes because protected nature areas are not located within city contours.

The two distance variables indicate that the further an urban greyfield plot is from a train station, the lower the chance of redevelopment, and this effect increases over time. Whereas this effect remains rather constant for non-urban greenfield development and is insignificant for the last to periods for urban densification. For the regional distance variable (travel time to 100.000 inhabitants), the odds ratio decreases over time implying that being further away from other people has a decreasing impact on the chance of development that grows stronger over time. Which means that development occurs more closer to more populous areas over time. Moreover, the results for the Urban Attractivity Index show odds ratios larger than 1 and slightly increasing over time for urban densification and urban greyfield. Which indicates the relevance of amenities, a higher level of amenities in the neighbourhood gives a higher chance of development and this increases over time and this is our sixth important conclusion. Moreover, the odds ratio for non-urban greenfield development slightly decreases but has a larger odd ratio than for the other processes. This could indicate that greenfield development only occurs at the city fringe.

### 4.3. Sensitivity

To test the sensitivity of the results for the choice of resolution we repeated the analysis at a finer (25 metre) resolution. This sensitivity test was performed for the third period for which the housing data could be prepared at the 25-metre resolution of the original land-use data. For the earlier periods the housing data is only available at a 100-metre resolution. The test shows almost identical results for the number of houses added in each process. The results are available upon request.

## 5. Conclusion and discussion

In this study, we investigated the effects of a shift in the spatial planning approach in the Netherlands with respect to the housing market. Country-wide coordination of housing construction via national spatially-explicit development zones faded out during the period 2000–2017 considered in this paper. The residential development zones that were designated areas at the start of the nineties and where a large share of new housing was realised between 1995 and 2005 [77] gradually became filled with new housing, while no new development zones were designated at the national level. Most other elements of the Dutch spatial planning system remained in place. We focussed on the evolution of residential densification and the redevelopment of greyfields and brownfield as residential areas over the period when the national planning policy was fading out. To study this, we collected detailed spatial housing and land-use data and considered three subsequent periods characterised by ever-decreasing national policy attention to housing. By doing so, we are able to study how the relative importance of different residential development processes changes over time. Our results confirm the trends observed in earlier studies on urban densification up to 2010 [52, 78] and extend them by decomposing to specific types of redevelopment processes: Additions to the housing stock in existing urban areas, notably greyfields, replaced greenfield development outside urban areas.

Establishing compact cities and fostering densification has been a key objective of national spatial planning since the 1980s. Our results for the Netherlands indicate that just over half of the total net increase of around one million residences in the past 18 years took place within the existing urban fabric of 2000. Interestingly enough, the total amount of new housing did not change substantially over time when the national construction programs on designated sites gradually faded out. While the national policy instruments got weaker, the share of houses added *within urban areas* strongly increased over time, thus compensating the decreasing greenfield development by various types of densification. This strength of this development is

even more noteworthy in light of the fact that we used urban contours of 2000 for all three 6-year time intervals studied, and one could expect that it became more difficult to add housing units when time progressed and the most obvious locations for (re)development were filled. Additionally, only a small proportion of the total increase in housing units was placed on former green areas within cities. Densification of existing residential areas continued at approximately the same pace over the whole period, while the transformation of former grey- and brownfields increased substantially and is the main reason behind the almost complete compensation of the loss of housing supply from greenfield development. Furthermore, the average density in housing units per hectare of those cells where houses were added outside urban areas is decreasing, while the density in urban greyfields is increasing. On top of that, it appears that the urban residential densification in more recent years was realised not by adding one or a few units more or less randomly over space, but through the realisation of larger numbers of dwellings at specific locations, for instance by replacing single-family housing by apartment buildings.

Our results lend further credence to the idea that the abandoned active national spatial planning policy targeting at housing construction on greenfield sites limited the need for intensifying the use of existing residential areas through redevelopment [52]. Despite reduced government spending on spatial planning aiming at dense urban development, densities increased within the existing city, as generic local-level restrictive policies with respect to greenfield development remained in place. The fading out of the national program of housing construction may have contributed to higher house prices within urban areas. Economic theory suggests that this stimulates the transformation of land used for other purposes to residential use, and this is consistent with the increasing importance of brownfield and greyfield development. Alternatively, the fading out of greenfield development programs may have induced intensification of plans for (re)development of housing in existing areas by the municipalities. It is somewhat ironic that the disappearance of the national program aiming at densification through building next to existing urban areas led to increased densification within urban areas with less greenfield development and roughly the same amount of housing construction. This development is even more surprising in light of the higher costs and efforts needed to realise urban redevelopment relative to greenfield development.

Consistent with the renewed interest in living in the city, or in other words, living in close proximity to a certain level of amenities, we found evidence for the relevance of proximity to amenities (as stipulated by, for example, Clark, Lloyd [70] and Glaeser, Gyourko [72]) and it is increasing over time. Furthermore, being further away from other people has a decreasing impact on the chance of development, and that grows stronger over time. Which means that development occurs closer to more populous areas over time.

The fading out of top-down planning shifted densification to a smaller scale, which tends to provide more room for bottom-up development initiatives in which the market plays a larger role [33]. Yet, the developments are also in line with current policy ambitions that since 2012 are formulated in non-spatially explicit policy guidelines. The shares of development inside urban contours we found may partly be explained by the implementation of this policy. This indicates that such non-spatially-explicit restriction strategies can be useful in enabling densification and preserving open space, and is consequently a tool in battling urban sprawl that can be recommended to policy makers The designation of large-scale residential development areas outside urban areas on the other hand could hinder densification efforts, because it acts as a valve that relieves demand pressure inside urban areas. Spatial planners should thus be careful in defining such development zones and be aware of its potential adverse implications for densification ambitions in existing urban areas.

The analysis of the present paper provides tangible insight into the effect of planning policies on the housing market. Urban densification will probably remain high on the policy agenda in the Netherlands, where urban house prices are still increasing and the availability of new housing locations is limited. Moreover, the findings presented here offer food for thought for policy makers in other countries that consider implementing new spatial policies or decentralisation of government responsibilities.

## Supporting information

**S1 Table. Aggregation of land-use classes.**
(DOCX)

**S2 Table. Actual development values per period for the three main development processes.**
(DOCX)

## Author Contributions

**Conceptualization:** Jip Claassens, Eric Koomen, Jan Rouwendal.

**Data curation:** Jip Claassens.

**Formal analysis:** Jip Claassens.

**Investigation:** Jip Claassens.

**Methodology:** Jip Claassens, Eric Koomen, Jan Rouwendal.

**Project administration:** Jip Claassens, Eric Koomen.

**Software:** Jip Claassens.

**Supervision:** Eric Koomen, Jan Rouwendal.

**Visualization:** Jip Claassens.

**Writing – original draft:** Jip Claassens, Eric Koomen.

**Writing – review & editing:** Jip Claassens, Eric Koomen, Jan Rouwendal.

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
