## [Decision Letter · Decision Letter 0]

17 Aug 2020

PONE-D-20-22192

Urban density and spatial planning: the unforeseen impacts of Dutch devolution

PLOS ONE

Dear Dr. Claassens,

Thank you for submitting your manuscript to PLOS ONE. After careful consideration, we feel that it has merit but does not fully meet PLOS ONE’s publication criteria as it currently stands. Therefore, we invite you to submit a revised version of the manuscript that addresses the points raised during the review process.

As the reviewers suggest, the submitted manuscript has good scientific grounds and generally the research is well structured and presented. However, there are minor issues as highlighted by the reviewers such as grammar corrections, missing information and minor structural changes. Therefore, I invite you to address the issues in the reviewer reports and re-submit the revised manuscript.

We look forward to receiving your revised manuscript.

Kind regards,

Eda Ustaoglu, PhD

Academic Editor

PLOS ONE

Journal Requirements:

"This research was financially supported by the Dutch Research Council (NWO) as a SURF Pop up project."

Reviewers' comments:

Reviewer's Responses to Questions

**Comments to the Author**

1. Is the manuscript technically sound, and do the data support the conclusions?

Reviewer #1: Yes

Reviewer #2: Yes

2. Has the statistical analysis been performed appropriately and rigorously? 

Reviewer #1: Yes

Reviewer #2: Yes

3. Have the authors made all data underlying the findings in their manuscript fully available?

Reviewer #1: Yes

Reviewer #2: Yes

4. Is the manuscript presented in an intelligible fashion and written in standard English?

Reviewer #1: Yes

Reviewer #2: Yes

5. Review Comments to the Author

Reviewer #1: Thank you for the opportunity to review the manuscript entitled “Urban density and spatial planning: the unforeseen impacts of Dutch devolution”. The authors investigated residential development since 2000 in the Netherlands in relation to the changing planning policies. The study found that while the national policy on residential development is decentralised making it look weak and the share of residential development within existing urban areas increased. The findings of this study contribute to literature especially its methodological approach. In general, this is a well-written manuscript. However, I have a few suggestions that when considered can help improve the manuscript.

Topic

1. The topic appears great because it is clear and specific to the purpose of the study.

Abstract

2. The sentences should be in the past. For instance, page 2 line 17 and 19 should be we focused… and we quantified… respectively.

3. Keywords: Please add rural women

Background

4. The background and problem statement looks great but kindly state the specific objectives.

5. Kindly add the significance of the study to the background.

Dutch Context

6. This section was well explained

Methods

7. The study design was not too clearly stated.

8. Please check the sentence of page 7 line 150, it should read …, we focused on the…

9. Is there any justification for putting brownfield redevelopment into bracket (line 177)?

10. Please check the sentence of line 181, it is supposed to read we use

Results and discussions

11. The results presented were great but the conclusion made could it be all summed in the conclusion and recommendation? If not, kindly state it as your key findings. For instance, line 300 “This is our first important conclusion: the disappearance of national coordination and the abolition of development subsidies did not result in a drop in the number of housing units added to the stock”. Similar of such conclusion could be found in line 309.

12. In some of the interpretations, please refer to the exact table number. For instance, line 303, “The table (Table 2) shows a large reduction in the share of new housing constructed outside existing urban areas: from 58% in the first period to 31% in the third”. This is similar in the following lines; line 323 and 373.

13. In line 372, the saldi stated here cannot be found in table 2. Kindly check.

14. Well synthesised, Good job done!

Conclusion and recommendation

15. This was done well but line 466-467 talks about proximity to certain amenities was not mentioned in the results and discussion so how do you draw conclusion on that?

Reviewer #2: The manuscript is technically strong and is a good scientific research with data that supports the conclusions. The conclusions is appropriately based on the data presented and the statistical analysis has been performed appropriately and the manuscript presented in an standard English. I would like to be able to offer more precise data on the evolution of housing prices and to establish clearer recommendations for public policy.

6. PLOS authors have the option to publish the peer review history of their article (what does this mean?). If published, this will include your full peer review and any attached files.

Reviewer #1: No

Reviewer #2: No

---

## [Author Response · Author response to Decision Letter 0]

14 Sep 2020

Reviewer #1

Thank you for the opportunity to review the manuscript entitled “Urban density and spatial planning: the unforeseen impacts of Dutch devolution”. The authors investigated residential development since 2000 in the Netherlands in relation to the changing planning policies. The study found that while the national policy on residential development is decentralised making it look weak and the share of residential development within existing urban areas increased. The findings of this study contribute to literature especially its methodological approach. In general, this is a well-written manuscript. However, I have a few suggestions that when considered can help improve the manuscript.

Thanks for your kind comments.

Topic

1. The topic appears great because it is clear and specific to the purpose of the study.

Thanks again.

Abstract

2. The sentences should be in the past. For instance, page 2 line 17 and 19 should be we focused… and we quantified… respectively.

We have changed the tense in the abstract and entire method section for consistency.

3. Keywords: Please add rural women

In relation to rural development we have listed the keyword greenfield redevelopment. The gender aspect of these developments is not discussed, so we have note included this suggestion. 

Background

4. The background and problem statement looks great but kindly state the specific objectives.

We have now emphasised the objectives in the introduction.

5. Kindly add the significance of the study to the background.

We have formulated the significance of the study in a more explicit way in the introduction section.

Dutch Context

6. This section was well explained

Thanks for this positive comment.

Methods

7. The study design was not too clearly stated.

We now sketch the basic study design at the start of the Methods section.

8. Please check the sentence of page 7 line 150, it should read …, we focused on the…

We have changed the tense throughout this section.

9. Is there any justification for putting brownfield redevelopment into bracket (line 177)?

In this research brownfield redevelopment is combined with greyfield redevelopment because of a lack of appropriate data, that has now been clarified in the caption of Fig 1 and the main text. 

10. Please check the sentence of line 181, it is supposed to read we use

The sentence has been adapted to the past tense and now reads “we used”.

Results and discussions

11. The results presented were great but the conclusion made could it be all summed in the conclusion and recommendation? If not, kindly state it as your key findings. For instance, line 300 “This is our first important conclusion: the disappearance of national coordination and the abolition of development subsidies did not result in a drop in the number of housing units added to the stock”. Similar of such conclusion could be found in line 309.

We have included more explicit conclusions to the results section and conclusion.

12. In some of the interpretations, please refer to the exact table number. For instance, line 303, “The table (Table 2) shows a large reduction in the share of new housing constructed outside existing urban areas: from 58% in the first period to 31% in the third”. This is similar in the following lines; line 323 and 373.

We have added the explicit table number in this sentence and also to two other instances in the Results section.

13. In line 372, the saldi stated here cannot be found in table 2. Kindly check.

We have described more explicitly now why the saldi here differ from Table 2.

14. Well synthesised, Good job done!

Thanks again for your kind words.

Conclusion and recommendation

15. This was done well but line 466-467 talks about proximity to certain amenities was not mentioned in the results and discussion so how do you draw conclusion on that?

We now more clearly indicated that the urban attractivity index represents a neighbourhood related amount of amenities.

Reviewer #2

The manuscript is technically strong and is a good scientific research with data that supports the conclusions. The conclusions is appropriately based on the data presented and the statistical analysis has been performed appropriately and the manuscript presented in an standard English. I would like to be able to offer more precise data on the evolution of housing prices and to establish clearer recommendations for public policy.

Thank you for this very positive review. In response we have now included a precise figure related to house price development in Section 4.1. 

In the conclusion we now introduced more specific public policy recommendations.

---

## [Decision Letter · Decision Letter 1]

2 Oct 2020

Urban density and spatial planning: the unforeseen impacts of Dutch devolution

PONE-D-20-22192R1

Dear Dr. Claassens,

We’re pleased to inform you that your manuscript has been judged scientifically suitable for publication and will be formally accepted for publication once it meets all outstanding technical requirements.

Kind regards,

Eda Ustaoglu, PhD

Academic Editor

PLOS ONE

Additional Editor Comments (optional):

Reviewers' comments:

Reviewer's Responses to Questions

**Comments to the Author**

1. If the authors have adequately addressed your comments raised in a previous round of review and you feel that this manuscript is now acceptable for publication, you may indicate that here to bypass the “Comments to the Author” section, enter your conflict of interest statement in the “Confidential to Editor” section, and submit your "Accept" recommendation.

Reviewer #1: All comments have been addressed

2. Is the manuscript technically sound, and do the data support the conclusions?

Reviewer #1: Yes

3. Has the statistical analysis been performed appropriately and rigorously? 

Reviewer #1: Yes

4. Have the authors made all data underlying the findings in their manuscript fully available?

Reviewer #1: Yes

5. Is the manuscript presented in an intelligible fashion and written in standard English?

Reviewer #1: Yes

6. Review Comments to the Author

Reviewer #1: Thank you for the opportunity to review the manuscript entitled “Urban density and spatial planning: the unforeseen impacts of Dutch devolution”. The authors investigated residential development since 2000 in the Netherlands in relation to the changing planning policies. The study found that while the national policy on residential development is decentralised making it look weak and the share of residential development within existing urban areas increased. The findings of this study contribute to literature especially its methodological approach. In general, this is a well-written manuscript. All concerns raised have been addressed.

7. PLOS authors have the option to publish the peer review history of their article (what does this mean?). If published, this will include your full peer review and any attached files.

Reviewer #1: No

---

## [Editor Report · Acceptance letter]

8 Oct 2020

PONE-D-20-22192R1 

Urban density and spatial planning: the unforeseen impacts of Dutch devolution 

Dear Dr. Claassens:

I'm pleased to inform you that your manuscript has been deemed suitable for publication in PLOS ONE. Congratulations! Your manuscript is now with our production department. 

Kind regards, 

on behalf of

Dr. Eda Ustaoglu 

Academic Editor

PLOS ONE